# ALDRESS: A Retrospective Pilot Study to Develop a Pharmacological Causality Algorithm for Drug Reaction with Eosinophilia and Systemic Symptoms (DRESS)

**DOI:** 10.3390/jcm13092622

**Published:** 2024-04-29

**Authors:** Stefan Stewart, Arturo Gómez López de las Huertas, María Jiménez-González, Antonio J. Carcas, Alberto M. Borobia, Elena Ramírez

**Affiliations:** 1Clinical Pharmacology Department, La Paz University Hospital-IdiPAZ, Faculty of Medicine, Universidad Autónoma de Madrid, 28046 Madrid, Spain; aglopezhuertas@salud.madrid.org (A.G.L.d.l.H.); antonio.carcas@uam.es (A.J.C.); alberto.borobia@salud.madrid.org (A.M.B.); 2Clinical Trial Unit, La Paz University Hospital-IdiPAZ, 28046 Madrid, Spain; maria.jimenez.gonzalez@salud.madrid.org

**Keywords:** adverse drug reactions, causality algorithms, drug reaction with eosinophilia and systemic symptoms (DRESS)

## Abstract

**Background:** The drug reaction with eosinophilia and systemic symptoms (DRESS) syndrome represents a severe form of drug hypersensitivity reaction characterized by significant morbidity, mortality, and long-term sequelae, coupled with limited therapeutic avenues. Accurate identification of the causative drug(s) is paramount for acute management, exploration of safe therapeutic alternatives, and prevention of future occurrences. However, the absence of a standardized diagnostic test and a specific causality algorithm tailored to DRESS poses a significant challenge in its clinical management. **Methods:** We conducted a retrospective case–control study involving 37 DRESS patients to validate a novel causality algorithm, the ALDRESS, designed explicitly for this syndrome, comparing it against the current standard algorithm, SEFV. **Results:** The ALDRESS algorithm showcased superior performance, exhibiting an 85.7% sensitivity and 93% specificity with comparable negative predictive values (80.6% vs. 97%). Notably, the ALDRESS algorithm yielded a substantially higher positive predictive value (75%) compared to SEFV (51.40%), achieving an overall accuracy rate of 92%. **Conclusions:** Our findings underscore the efficacy of the ALDRESS algorithm in accurately attributing causality to drugs implicated in DRESS syndrome. However, further validation studies involving larger, diverse cohorts are warranted to consolidate its clinical utility and broaden its applicability. This study lays the groundwork for a refined causality assessment tool, promising advancements in the diagnosis and management of DRESS syndrome.

## 1. Introduction

Drug reaction with eosinophilia and systemic symptoms (DRESS) syndrome is a potentially severe type of drug hypersensitivity reaction associated with high morbidity and mortality, long-term effects, and limited therapeutic options [1]. Its prevalence is estimated to be between 2.18 and 9.63 per 100,000 people [2,3]. The growing use of targeted anticancer drugs and immunotherapies may further raise this rate in the coming years. As a result, it is becoming essential for dermatologists to be proficient in the diagnosis and treatment of DRESS [4].

The disease’s clinical immunological mechanism is defined by drug intake or introduction, with symptoms typically appearing two to eight weeks after administration. It is distinguished by the appearance of confluent maculopapular erythematous lesions predominantly on the trunk, face, limbs, and arms, usually preceded by a febrile illness in the previous days. It initially appears on the face and upper trunk before spreading downwards. During the early stages, the most distinctive cutaneous symptoms are periorbital and facial edema with pinhead-sized pustules [5]. If the medications causing the illness are not discontinued, the rash may progress to severe exfoliative dermatitis or erythroderma. Blisters may occasionally appear but are usually limited to the wrists and are likely caused by cutaneous edema. In rare cases, there may be lesions on mucosal surfaces [5].

Systemic impact varies widely. Nearly 90% of patients experience involvement of at least one internal organ, with around 35% facing issues with two organs, and up to 20% experiencing the involvement of more than two organs [1,3]. In some cases, visceral manifestations might arise before the typical rash associated with the condition [3]. Liver complications affect 53 to 90% of cases [2]. Most abnormalities in liver function tests are temporary and relatively mild and include a wide array of liver injury patterns such as cholestatic, hepatocellular, and mixed types [1]. Although rare, acute liver failure can occur in DRESS patients and may lead to liver transplant. Kidney issues range from proteinuria to renal failure, with acute interstitial nephritis as a common finding [2]. Patients may experience acute renal failure, requiring dialysis in about 3% of cases. Elderly patients or those with pre-existing renal or cardiovascular conditions are at higher risk, particularly if the reaction is linked to allopurinol [1]. Pulmonary complications affect up to 30% of patients, often causing shortness of breath and a dry cough. Imaging studies may reveal interstitial infiltrates and pleural effusions, with potential complications including pneumonitis and acute respiratory distress syndrome [5]. Cardiac involvement, occurring in 2 to 20% of cases, is a severe complication. Symptoms may include hypotension, tachycardia, and chest pain, with various types of myocarditis presenting different prognoses. Hypersensitivity myocarditis generally carries a better outlook, while eosinophilic myocarditis is associated with high mortality [2]. Other organ systems can also be affected, albeit less commonly. Central and peripheral nervous system issues, gastrointestinal complications, and rare complications like myositis and thyroid dysfunction have been reported, underlining the systemic nature and potential severity of DRESS syndrome [2,3].

Alterations in the hemogram often include absolute and relative eosinophilia elevation, although this is not the solely finding, as atypical lymphocytes are also frequently observed [1]. Serum IgG, IgA, and IgM levels are frequently reduced during the acute stage, but they eventually return to normal following full recovery [6]. The severity of the condition depends on the extent of the systemic involvement, with the most prominent alteration being that of the liver profile, followed by the renal and pancreatic profiles, although rhabdomyolysis, myocarditis, or pneumonitis may also be seen [1].

DRESS is a delayed-type drug hypersensitivity reaction (DHR) mediated by T cells. DRESS’s immunological mechanism is defined as a type IV hypersensitivity reaction, which is characterized by lymphocyte activation, including CD4+ and CD8+ T cells. By detecting the activation and expansion of drug-specific memory T cells to the suspected implicated drug, the lymphocyte transformation test (LTT) is used to diagnose drug-induced hypersensitivity reactions. According to the ENDA/EAACI Drug Allergy Interest Group, an LTT is recommended before in vivo tests in severe reactions with a suspected T cell mechanism. Using the algorithm of the Spanish pharmacovigilance system (SEFV) as the gold standard, 82% of DRESS cases were associated with a variety of drugs when this test was performed after recovery [7].

Studies focused on pharmacogenomics, provided evidence that some HLA alleles predispose to DRESS and other delayed-type DHRs to certain drugs [8]. Particular HLAs, such as HLA-B*15:02 and HLA-A*31:01 for new carbamazepine users, HLA-B*58:01 for allopurinol, and HLA-B*57:01 for abacavir, have been successfully used in clinical praxis to safeguard patients from the development of DRESS [9]. Also, several genetic predictors for T cell-mediated hypersensitivity reactions to CBZ have been identified, including TNF2, HLA-DR3, HLA-DQ2, HSP70, HLA-B*15:02, HLA-A*31:01, HLA-B*59:01, HLA-B*15:11, HLA-A*24:02, HLA-B*40:01, and HLA-B*51:01 [10]. Furthermore, it has been observed that HLA-A*32:01 is highly related with vancomycin-induced DRESS in a group of largely European descent [11]. Thereby, testing for HLA haplotypes could improve drug safety, help identify the causative medication, and protect future drug therapy alternatives. Nevertheless, implementing widespread HLA testing can be complex due to limited knowledge of HLA allele associations with drugs, and varying prevalence of at-risk HLA alleles among different populations [9]. 

On the other hand, viral infection or its reactivation is also considered a contributing factor in DRESS. Clinical evidence has demonstrated reactivation of human herpesvirus-6 and 7, Epstein–Barr virus, and cytomegalovirus in DRESS [12]. Systemic viral infections typically precede drug reactivity, which can be stimulated in two ways. Firstly, it occurs through virus-induced second signals. For example, certain drugs like β-lactam antibiotics act as haptens and covalently bind to various soluble and tissue proteins, forming new antigens [13]. These neoantigens do not usually induce an immune reaction under normal conditions, likely due to a lack of co-stimulation [14]. However, during a viral infection, hapten-modified peptides are presented in an immune-stimulatory environment under co-stimulation. This may lead to the development of a drug-specific immune reaction, which can appear as an exanthema [13]. Secondly, drugs have a tendency to bind simultaneously to proteins and immune receptors (p-i). Although this low-affinity binding may be ineffective to induce T cell activation in the absence of a viral infection [15], during a viral infection, immune receptors are more highly expressed and provide for more interactions. Thus, the probability for increased low affinity interactions increase and then, drug p-i binding may become functionally relevant. T cells are activated by the drug p-i as long as the virus-induced immune stimulation lasts [12,16]. Alternatively, DRESS-induced hypersensitivity can also occur initially, followed by viraemia of endogenous herpes viruses such as human herpesvirus 6 (HHV6), cytomegalovirus (CMV), and Epstein–Barr virus (EBV) [17]. This viraemia may appear acutely, but is often detected 3–6 weeks later. In most cases, viraemia develops after stopping the medication that caused DRESS [18]. It has been hypothesized that this may occur due to massive p-i mediated immune stimulation during acute DRESS, which can activate many herpes virus-specific T cells. The p-i stimulation affects both naïve and memory T cells, causing activation and expansion of various T cells, including polyclonal, polyspecific, and cytotoxic T cells. Within this pool of p-i activated T lymphocytes are T cells that are specific for endogenous herpes viruses which control viral replication, likely via local IFN-γ release. When these T cells encounter their target stimulus (i.e., HHV6, CMV, EBV) in peripheral tissue, they kill the herpes-infected cells and as a consequence, endogenous intracellular herpesviruses are released [12].

Identifying the causal drug or drugs is crucial in preventing future reactions and identifying safe therapeutic alternatives. To correctly identify the culprit drug, an accurate diagnosis of the clinical entity is necessary, including the time window from the first drug intake to the onset of the disease and the suspected involvement of infectious agents or alternative causes. However, this can be challenging, particularly when multiple drugs are being used concurrently [7]. Additionally, DRESS cases often involve sensitization to multiple drugs, including those administered during the acute reaction [7]. In the absence of a gold standard diagnosis methodology, the approach for conducting a causal assessment of adverse drug reactions relies heavily on the causality algorithms tool [19]. The use of these algorithms ensures objectivity and comprehensibility in the assessment process. These algorithms offer structured and standardized assessment methods, enabling clinicians to systematically identify ADRs based on parameters such as time to onset of adverse events, previous drug/adverse reaction history, and triggering and re-exposure. The algorithms for causality may exhibit a high sensitivity, with a positive predictive value, but sometimes low specificity and negative predictive value. They also demonstrate high levels of inter-observer agreement beyond chance [19]. Spanish-speaking countries typically use the algorithm developed by the Spanish Pharmacovigilance System (SEFV) [1]. All of these algorithms are based on the Karch and Lasagne algorithm and differ in their sets of questions and associated scores used to calculate the probability of a cause–effect relationship [19]. The final evaluation of each medication is classified as unrelated (if not classified, unlikely, or conditional) or related (if possible, probable, or definite). Currently, the ALDEN score is recognized as one of the most reliable tools for identifying culprit drugs in SJS and TEN [20,21]. It is generally used for retrospective evaluation of drug causality, but it may also be useful in the acute phase of the disease when the yield of allergy diagnostic testing is low. Nevertheless, in contrast to SSJ/TEN, there is currently no validated scale for establishing pharmacological causality in DRESS. This results in the application of non-specific causality algorithms that often result in broad pharmacological restrictions, limiting the patient’s therapeutic arsenal for the future. 

Nowadays, the diagnosis of DRESS is based on specific criteria developed by a consortium group known as the European Registry on Severe Cutaneous Adverse Drug Reactions (RegiSCAR) [22]. The core clinical manifestations include fever, enlarged lymph nodes, hematological findings such as eosinophilia and the presence of atypical lymphocytes, as well as cutaneous and visceral involvement, with time to resolution and evaluation of alternative causes also taken into account [22,23]. It defines four levels of certainty: excluded, possible, probable, and definitive. Scores between +2 and +3 indicate a possible DRESS syndrome, while a score of +4 indicates probable, and a score of +5 to +8 indicates definitive [22]. It should be noted that the RegisCAR scale does not include reactivation of HHV6 among its criteria, despite its association with DRESS. However, other scales, such as the Japanese consensus group’s do consider it [18,24]. The Spanish multidisciplinary and multicentric researching consortium on SCARs, named PIELenRed, which was created in 2010, was further integrated into RegiSCAR and contributes greatly to provide reliable epidemiologic data of DRESS in Spain [1]. 

The Hospital Universitario la Paz Clinical Pharmacology Service operates a Hospital Pharmacovigilance Programme by Laboratory Signals (PFVHSL). The program proactively detects and diagnoses eosinophilia that are compatible with DRESS conditions and that cause or occur during hospitalization. The PFVHSL is based on the fact that certain altered analytical parameters may indicate a serious adverse reaction. The Pharmacovigilance Section of the Clinical Pharmacology Service is a member of PIELenRed, a multi-center, multidisciplinary consortium dedicated to the study of serious adverse skin reactions to drugs and a member of RegiSCAR. This platform gathers the drug reactive cases detected in hospitals in the Community of Madrid and is responsible, among other functions, for collecting information on reported cases, which are processed and included in a database; follow-up of patients for up to 1 year and 5 years after being discharged; maintaining a biobank of biological samples; estimating specific incidences of severe cutaneous reactions for each of the suspected drugs; assessing the incidence of each of the suspected drugs; evaluating the various diagnostic tests; and quantifying the association between the most frequent alleles of HLA-I and HLA-II in patients who develop cutaneous adverse reactions [25]. The Hospital Universitario La Paz offers a range of medical services, including allergology, dermatology, intensive care, plastic surgery, and clinical pharmacology. These services in particular are grouped under the PFVHSL, which is a part of PIELenRed. Since the establishment of PFVHSL in 2007, they have detected a total of 37 cases of potential DRESS. However, PIELenRed has identified instances of DRESS in other hospitals within the consortium. The main objectives were to validate an algorithm for causality assessment of drug reaction with eosinophilia and systemic symptoms (DRESS) syndrome (ALDRESS), to assess the inter-observer agreement of ALDRESS, and to compare ALDRESS with the SEFV algorithm.

## 2. Materials and Methods

### 2.1. Design

This is an observational retrospective case–control study. The protocol was approved by the Ethics Committee of La Paz University Hospital and validated by the PIELenRed Expert Committee.

### 2.2. Selection of Cases and Controls

The selection was made from the PFVHSL database as well as from the PIELenRed database. Inclusion criteria was a RegiSCAR scale score for DRESS diagnosis greater than +2 due to a single drug, and all DRESS cases with a score greater than +2 due to more than one drug (Appendix A). Cases included patients with DRESS where the culprit drug was the only drug administered. The control group in the studies comprised patients for whom more than one drug was under suspicion as a potential cause, with at least one of these drugs being identified as a culprit in the case group. 

### 2.3. Data Collection Procedure

The pharmacological history was obtained from the electronic health record through the HCIS program and from the data collection notebook of the cases included in PIElenRed. All participants were assessed using both the ALDRESS algorithm and the Spanish Pharmacovigilance System algorithm (SEFV). Two independent researchers conducted the evaluations in a blinded fashion. Both algorithms (ALDRESS and SEFV) were evaluated by two independent researchers. 

The ALDRESS algorithm includes demographic data, such as age, sex, co-morbidities, and risk factors, REGISCAR score > 2, along with the index date and resolution date, medication information (drug: type, start and stop times, notoriety, challenge, rechallenge, and concomitant drugs), microbiological examination to detect the presence of the herpes virus, as well as other viruses and bacteria such as HHV Type B/C and mycobacteria, and immunological evaluation was conducted using the LTT Test and epicutaneus allergy tests (Appendix A). 

The SEFV algorithm assigns a score to each of seven factors that are considered to be relevant in determining whether an adverse drug reaction (ADR) is causally related to a medication. The factors are chronology, literature, withdrawal, rechallenge, alternative causes, contributing factors, and complementary explorations. The chronology was considered compatible between 1 and 7 days or up to 21 days later. In rare cases, symptoms can appear up to 21 days later, but it was considered not fully compatible (Appendix A).

### 2.4. Data Analysis

A descriptive analysis per suspected drugs was conducted. A descriptive analysis of the clinical characteristics of the subjects was carried out. The frequencies of the distinct clinical parameters were determined using the SPSS 26 statistical program. The sensitivity, specificity, and predictive values of the ALDRESS or SEFV algorithms for both groups were calculated and compared using Fisher’s exact test using R (Integrated Development for R. RStudio, PBC, Boston, MA; http://www.rstudio.com, accessed on 31 January 2024).

Additionally, inter-observer variability was assessed using R. The degree of agreement between the two observers was calculated using Cohen’s Kappa index and the values obtained were interpreted according to: (0) no agreement; (0.10–0.20), slight agreement; (0.21–0.40) fair agreement; (0.41–0.60) moderate agreement; (0.61–0.80) substantial agreement; (0.81–0.99) near perfect agreement [26].

## 3. Results

### 3.1. Population and Clinical Evaluation

A total of 37 patients with a median age of 41.62 ± 17.09 years were evaluated. The population was balanced in sex, with the proportion of men (54.1%) being slightly higher than that of women (45.9%). Of the 37 patients evaluated, 7 (8.10%) were cases where the culprit drug was the only drug administered, and 30 were controls (81.08%) with more than one drug administered. The mean time to onset was of 23 days.

The most frequently administered drugs were lamotrigine, which corresponds to a synthetic phenyltriazine. Amoxicillin/clavulanic acid and carbamazepine were also frequently administered (Table 1). 

Regarding the antecedents of autoimmune conditions, it was observed that in this group, 89.1% (33/37) of the patients had no records or did not know if they suffered from any autoimmune disease. Of the remaining patients, 5.4% (2/37) had antecedents of rheumatoid arthritis, 5.4% (2/37) psoriatic arthritis, and 2.7% (1/37) of ankylosing spondylitis.

### 3.2. Microbiological Tests 

The data from microbiology tests (immunoassay or PCR) were analyzed and the frequency of positive and negative cases as well as those without any record were calculated. It was found that for mycobacteria 5.4% (2/37) of the patients had a positive result, 5.4% (2/37) had negative results, and 89.2% (33/37) did not have any record of this test. For EBV, 10.8% (4/37) were positive, 32.4% (12/37) were negative, and 56.8% (21/37) had no record. For HHV-6, it was found that 45.9% (17/37) were negative, 54.1% (20/37) had no record, and there were no patients with positive results. For CMV, 18.9% (7/37) were positive, 37.8% (14/37) were negative, and 43.2% (16/37) had no records. For VHC, 43.2% (16/37) were negative and 56.7% (21/37) had no record. For HIV, 45.9% (17/37) were negative and 54% (20/37) had no record. For parvovirus, 2.7% (1/37) of the patients were positive, 35.1% (13/37) were negative, and 62.1% (23/37) had no record. For VHS, 10.8% (4/37) were positive, 18.9% (7/37) of the patients were negative, and 70.2% (26/37) had no record. For VVZ, 5.4% (2/37) of the patients had positive results, 5.4% (2/37) had negative results, and 89.1% (33/37) had no records.

### 3.3. Immunological Tests

The lymphocyte transformation test (LTT) as well as epicutaneous allergic tests were conducted, with the results shown in Figure 1. The majority of patients, 45.9% (17/37) had negative results for the LTT, and a significant proportion of patients, 35.1% (13/37) did not undergo the test. Only three patients had a positive LTT, two of whom also had positive epicutaneous allergic tests. 

### 3.4. Clinician Consensus When Compared to ALDRESS or SEFV Algorithm 

The ALDRESS algorithm had a high sensitivity (85.7%) and specificity (93.3%) in predicting causality for DRESS. It also had a positive predictive value of 75% and a negative predictive value of 96.6% (Table 2). The accuracy was independently calculated and found to be 91.9%. Similarly, sensitivity, specificity, PPV, and NPV of clinical consensus when using SEFV algorithm was calculated. Results are described in Table 3. When assessing the sample population with the SEFV algorithm, results yielded a positive predictive value of 51.4% and a negative predictive value of 91.00%, with overall lower sensitivity and specificity scores.

Comparisons of the results obtained using the distinct algorithms are shown in Figure 2. The ALDRESS algorithm was the most accurate globally, achieving a success rate of 92%. In terms of sensitivity and specificity, the SEFV algorithm had a confirmation function (sensitivity of 78%), while the ALDRESS algorithm had a discard function (specificity 93%) for the DRESS. The negative predictive values for both algorithms were similar (96.55% vs. 91%). However, the ALDRESS algorithm had a higher positive predictive value (75%) compared to the SEFV algorithm (51.40%).

### 3.5. Degree of Agreement between Test Values and Inter-Observer Variability

In order to adequately assess inter-observer variability, the Kappa index for agreement between the ALDRESS and SEFV algorithms was performed by two independent investigators who assessed study population with the ALDRESS and SEFV algorithms, without knowing what the clinical consensus on those subjects was. Agreement among both algorithms was moderate (k = 0.576), with a 95% confidence interval of 0.32 to 0.83 (Table 4).

Table 5 and Table 6 show the level of agreement as showcased by the Kappa index when applying the ALDRESS and SEFV algorithms, respectively.

The Kappa index for agreement between the two observers regarding the ALDRESS algorithm was 0.88, with a 95% confidence interval (CI) of 0.77 to 0.99. Thus, according to Landis & Koch, (1977) [26] the agreement between the two researchers was near perfect.

It can be seen in Table 6 that the Kappa index for agreement between the two researchers regarding the SEFV algorithm was 0.88, with a 95% confidence interval of 0.80 to 0.97. Similar to the data observed in Table 5, the agreement between the two observers was near perfect.

## 4. Discussion

In this retrospective study, a new causality algorithm developed for the DRESS entity was assessed and compared to the SEFV algorithm currently used to establish pharmacological causality in our setting. Validating this new algorithm (ALDRESS) would help to prevent future relapses and guide the selection of appropriate therapeutic options for each patient. Data from 37 patients with DRESS was retrospectively recorded. It was found that the most frequent culprit drug was lamotrigine followed by amoxicillin/clavulanic acid and carbamazepine. These findings are consistent with previous reports, which have identified carbamazepine and lamotrigine as two of the most frequent causes of DRESS [27,28,29,30]. On the other hand, the reported association between the use of common antibiotics, particularly in the pediatric population, and the occurrence of DRESS is concerning [31,32], where the majority of the incidence of DRESS was associated with amoxicillin/clavulanic acid. This situation could potentially lead to an increase in the incidence of DRESS due to the widespread use of this antibiotic in the population.

Because DRESS syndrome is associated with considerable morbidity and mortality, proper care is critical. Once the diagnosis of DRESS syndrome has been established, the next step in treatment is to discontinue the causative medication(s) immediately. The WHO global introspection method, despite its usefulness, has been subject to criticisms of subjectivity and imprecision since it is mainly based on expert clinical judgments [33]. Thereby, a proper algorithm to identify drug causality is a pivotal tool to further treatment.

The RegiSCAR algorithm has been shown to have high sensitivity and specificity for the diagnosis of DRESS. The algorithm was developed and validated using a large cohort of patients with DRESS. It is a valuable tool for clinicians who are evaluating patients with suspected DRESS [34]. Another algorithm, developed by Choudhary et al. [35], includes thymus and activation-regulated chemokine (TARC) level (>613.25 pg/mL), total body surface area (TBSA, >35%), high-sensitivity C-reactive protein (hsCRP, >5 mg/L), eosinophils (>6%), absolute eosinophil count (>450 cells/mm^3^), and aspartate transaminase (>92 U/L). The effectiveness of this algorithm in diagnosing the DRESS syndrome was statistically similar to the effectiveness of the RegiSCAR DRESS validation score (≥2). The combination model TBSA at baseline, eosinophil count, and hsCRP (denominated as CET 9) at the cutoff of 6.8 had a sensitivity of 96% and a specificity of 100% [34]. Further, the effectiveness of this algorithm was confirmed, showing a sensitivity of 94.29%, a specificity of 60%, a positive predictive value (PPV) of 80.5%, and a negative predictive value (NPV) of 85.7% [36]. 

Even though this is a preliminary test, the ALDRESS algorithm showed potential to adequately assess pharmacological causality in patients with RegiSCAR ≥ 2 with a high level of specificity and sensitivity, thus bridging the gap of the unmet need of a DRESS-specific clinical assessment tool that adequately attributes drug causality. Thus, in 86.49% of cases, the application of ALDRESS algorithm by two independent observers resulted in a matching net score which was similar to the matching observed (89.19%) with the application of SEFV algorithm under similar conditions (two independent observers). 

Besides the algorithm defined by the SEFV to determine causality in DRESS [36], there is no existing literature on a particular algorithm for establishing causality. Nevertheless, some research has linked causality to specific DRESS symptoms. For instance, a study analyzed the diversity of DRESS patterns based on the culprit drugs in cases reported to the Department of Clinical Pharmacology at Monastir University Hospital over a 15-year period. DRESS was most commonly caused by anticonvulsant drugs (27%), followed by allopurinol (26.3%) and antibiotics (24%). Anticonvulsant drugs were linked to increased lymphadenopathy, less renal involvement, and more positive skin tests. The allopurinol group had a higher mean age and a lower incidence of lymphadenopathy and renal damage. Antibiotics resulted in a lower rate of eosinophilia, shorter recovery time, and a lower RegiSCAR score. Allopurinol has been linked to severe kidney damage, whereas antibiotics have a short delay and a low RegiSCAR score [37]. In another study, the relationship between antiviral medications and DRESS was examined using data mining algorithms such as proportional reporting ratio (PRR) (≥2) with associated χ^2^ value (>4), reporting odds ratio (ROR) (≥2) with 95% confidence interval, and case count (≥3). Furthermore, molecular dosimetry studies were performed to investigate the interactions between selected antiviral drugs and potential targets. Among the antiviral agents selected, a potential DRESS diagnostic was associated with abacavir, acyclovir, ganciclovir, lamivudine, lopinavir, nevirapine, ribavirin, ritonavir, and zidovudine [38].

In the current study, the discrepancies in net scores for ALDRESS algorithm varied from two to four points according to the report of the two observers, and one to two points for SEFV algorithm. The most common reason for differing ALDRESS scores was difficulty in precise scoring in cases where concomitant medication had a strong association with DRESS. Other differences in ALDRESS net scores were attributed to variations in notoriety scores and subjective interpretation of partially compatible chronology.

Another factor influencing the results is related to the great proportion of patients with DRESS diagnosis and negative LLT results. Also, there was an important proportion of the patients with incomplete data. Likewise, disparities in SEFV algorithm net scores were also found to be caused by similar challenges, including differences in notoriety scores, subjective interpretation of partially compatible chronology, and incomplete data.

Differences in ALDRESS scoring led to differing overall causality categories in 13.51% and were more frequent at lower scores. Similar results were obtained for SEFV scoring. These results were confirmed by the results obtained in the reliability testing carried out using the Cohen’s Kappa indicator. Thereby, both algorithms (ALDRESS and SEFV) are suitable for DRESS diagnostic evaluation. Nevertheless, the algorithm’s accurate use must be based on reliable data that reflect the use of all the necessary tests, so the hospital’s registration system must be optimized, and emphasis must be placed on the complete evaluation of the parameters specified in the algorithm.

As far as we know, ALDRESS is the first specific algorithm for the DRESS entity. Even though this is a preliminary test, the ALDRESS algorithm showed potential to adequately assess pharmacological causality in patients with RegiSCAR ≥ 3 with a high level of specificity and sensitivity. Thus, in 86.5% of cases, the application of ALDRESS algorithm by two independent observers resulted in a matching net score which was similar to the matching observed (89.2%) with the application of SEFV algorithm under similar conditions (two independent observers). 

SEFV algorithm had both lower sensitivity and specificity rates when compared to the ALDRESS algorithm. 

### Strengths and Limitations of the Study

To the best of our knowledge, this study is the first of its kind to assess a novel, DRESS-specific, causality algorithm. This study, as well as the novel specific algorithm it evaluates, have been preliminarily validated using a retrospective case–control population. Due to the lack of a gold-standard when assessing drug causality in DRESS, we resorted to evaluating the algorithm on patients who were receiving only one drug when they presented with DRESS. Another strength of the ALDRESS algorithm was the low inter-observer variability, showcasing a higher objectivity when compared to its SEFV counterpart. Moreover, all assessed test metrics were superior when applying the ALDRESS algorithm when compared to the SEFV algorithm.

It should be noted that this study is not without limitation that can potentially hinder its results. First and foremost, it should be acknowledged that among both cases and controls, DRESS was defined as RegiSCAR score ≥ 2, and while often utilized as DRESS, it does not definitively establish the diagnosis. Reevaluation of the algorithm validity is warranted if a gold-standard diagnostic tool arises for DRESS.

Furthermore, the retrospective design and limited sample size introduce potential biases inherent to any retrospective analysis that could compromise the results obtained. The absence of a universally accepted gold standard for DRESS diagnosis coupled with the previous challenge, amplifies concerns regarding subjectivity and the generalizability of findings, particularly given ALDRESS’s reliance on expert opinion. Additionally, the incomplete availability of patient medical records in the hospital’s databases poses a notable hurdle to any comprehensive evaluation carried out retrospectively.

## 5. Conclusions

In conclusion, while both the ALDRESS and SEFV algorithms demonstrate promise in DRESS diagnostic assessment, ALDRESS emerges as globally more accurate. Nonetheless, the inherent ambiguity of RegiSCAR ≥ 2 underscores the complexity of DRESS diagnosis. Addressing this limitation and optimizing data acquisition processes are crucial for enhancing the clinical utility of ALDRESS and ensuring its effective implementation in diverse patient populations. Further efforts should be geared toward expanding on the validation of the algorithm by analyzing a more extensive dataset.

## Figures and Tables

**Figure 1 jcm-13-02622-f001:**
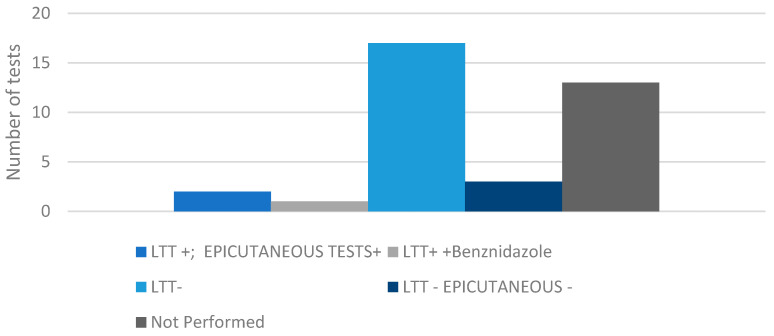
Immunological evaluation.

**Figure 2 jcm-13-02622-f002:**
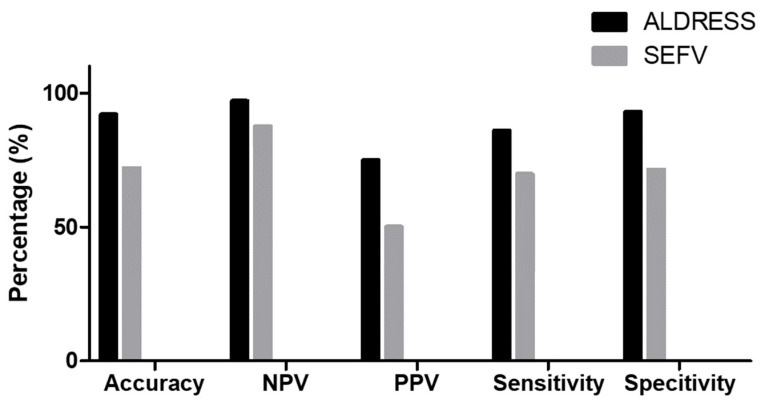
Comparison of clinician consensus of DRESS positive diagnosis using the ALDRESS or SEFV algorithm.

**Table 1 jcm-13-02622-t001:** Frequency of suspected medication administered before appearance of DRESS symptoms.

Suspected Medication	N	Percentage (%)
**Amoxicillin**	4	10.8
**Amoxicillin—Clavulanic Acid**	7	18.9
**Benznidazole**	1	2.7
**Carbamazepine**	7	18.9
**Cotrimoxazole**	5	13.5
**Eslicarbazepine**	1	2.7
**Lamotrigine**	9	24.3
**Phenytoin**	1	2.7
**Sulfasalazine**	2	5.5
**Total**	37	100.0

The mean time to onset was 23 days.

**Table 2 jcm-13-02622-t002:** Sensitivity, specificity, and predictive values of clinical consensus when using ALDRESS. ALDRESS positivity was defined as a score >7.

Metric	Value	95% Confidence Interval
**Sensitivity**	0.78	0.4321 to 0.9341
**Specificity**	0.75	0.3498 to 0.9802
**Positive Predictive Value**	0.514	0.2841 to 0.7010
**Negative Predictive Value**	0.91	0.5398 to 0.9881
**Likelihood Ratio**	3.879	

**Table 3 jcm-13-02622-t003:** Sensitivity, specificity, and predictive values of clinical consensus when using SEFV. SEFV positivity was defined as a score >6.

Metric	Value	95% Confidence Interval
**Sensitivity**	0.8571	0.4213 to 0.9964
**Specificity**	0.9333	0.7793 to 0.9918
**Positive Predictive Value**	0.7500	0.3491 to 0.9681
**Negative Predictive Value**	0.9655	0.8224 to 0.9991
**Likelihood Ratio**	12.86	

**Table 4 jcm-13-02622-t004:** Agreement between algorithms.

Kappa (k)	Value	ASE	Z	Pr (<|z|)	95% CI
**Unweighted**	0.5761	0.1307	4.409	1.038 × 10^−5^	0.320–0.832
**Weighted**	0.5761	0.1307	4.409	1.038 × 10^−5^	0.320–0.832

**Table 5 jcm-13-02622-t005:** Kappa index level of agreement between observers for ALDRESS.

Kappa (k)	Value	ASE	Z	Pr (<|z|)	95% CI
**Unweighted**	0.7926	0.08291	9.56	1.182 × 10^−21^	0.630–0.955
**Weighted**	0.8796	0.05454	16.13	1.658 × 10^−58^	0.772–0.985

**Table 6 jcm-13-02622-t006:** Kappa index level of agreement between observers for SEFV algorithm.

Kappa (k)	Value	ASE	Z	Pr (<|z|)	95% CI
**Unweighted**	0.7633	0.08386	9.102	8.854 × 10^−20^	0.598–0.927
**Weighted**	0.8847	0. 04407	20.075	1.227 × 10^−89^	0.798–0.971

## Data Availability

Data are contained within the article.

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
