# Peer review of "ALDRESS: A Retrospective Pilot Study to Develop a Pharmacological Causality Algorithm for Drug Reaction with Eosinophilia and Systemic Symptoms (DRESS)"

_jcm, 2024, doi:10.3390/jcm13092622_

Round 1
Reviewer 1 Report
Comments and Suggestions for Authors
The authors have chosen a very complex topic which is at the same time very relevant.
The presented new algorithm is based on a very small number of analyzed patients, but the results appear convincing.
What would definitely help the future introduction of this modality - and thus support its relevance - would be for the authors to suggest whether, and ideally how, they intend to resolve the current apparent intricacy of the process so that it can be implemented in routine practice.
Author Response
Esteemed reviewer 1
We would like to express our gratitude for your insightful comments on our manuscript. We appreciate the opportunity to address your concerns regarding the implementation of our algorithm in routine clinical practice.
We acknowledge that the sample size of patients analysed in our study is relatively small, which is to be expected given the low prevalence of DRESS, more so when it was required per protocol to include subjects with the disease that were had only one suspected drug as culprit. However, we are encouraged by the positive results obtained from our algorithm, which demonstrate its potential utility.
In response to your suggestion, we agree that providing clarity on the process's implementation in routine practice is crucial for its relevance and adoption. However although the validation process was complex, routine application of the ALDRESS algorithm should be straightforward, as it does not require any specific training.
However, it should be noted that we plan to focus our future research efforts on several key areas:
Validation and Expansion: We intend to conduct further validation studies on larger patient cohorts to strengthen the robustness and generalizability of our algorithm. By analyzing a more extensive dataset, we aim to better understand its performance across diverse patient populations and clinical settings.
Optimization of Implementation: We recognize the importance of streamlining the implementation process to facilitate its integration into routine clinical workflows. This involves refining the algorithm's user interface, optimizing computational efficiency, and addressing any logistical challenges associated with its deployment.
Stakeholder Engagement: We are committed to fostering collaboration with clinicians, healthcare providers, and other relevant stakeholders to ensure the seamless adoption of our algorithm into clinical practice. By actively engaging with end-users, we aim to solicit feedback, address concerns, and identify opportunities for improvement.
Clinical Guidelines and Protocols: We plan to develop comprehensive guidelines and protocols outlining the step-by-step procedures for utilizing our algorithm in clinical practice. These resources will serve as valuable tools for healthcare professionals, facilitating the standardized implementation and utilization of our modality.
In summary, we recognize the importance of addressing the current apparent intricacy of the implementation process to enhance the practical relevance of our algorithm. Through ongoing research, optimization efforts, and stakeholder engagement, we are confident in our ability to overcome these challenges and facilitate the widespread adoption of our modality in clinical practice.
Regards,
Dr. S Stewart
Dr. E Ramírez

Reviewer 2 Report
Comments and Suggestions for Authors
This study is a retrospective study of 37 patients with possible, probable and definite DRESS syndrome. The authors have not included in their introduction or in analysis visceral manifestations of the syndrome such as renal, pulmonary, gastrointestinal, cardiac etc. In fact, visceral involvement is one of the most prominent DRESS features in severe cases. As such, a conclusion regarding ALLDRESS algorithm in its current form cannot be made. The only way I can see this paper to be published is to make this as a retrospective study of 37 cases. Additionally, even if the methodology was sounds ( which is not) num,ber of cases is too small to develop an algorythm. References are old and none of the recent papers ( including the ones published in JCM) were not reviewed.
Comments on the Quality of English Languageminor edits
Author Response
Esteemed reviewer 2,
Thank you for your thorough evaluation of our manuscript and for providing valuable feedback. We appreciate the opportunity to address the concerns raised regarding the retrospective nature of our study, the inclusion of visceral manifestations, the methodology, and the adequacy of the literature review.
Visceral manifestations and introduction:
We acknowledge the need to address in the introduction the visceral manifestations of DRESS syndrome and such appropriate modifications have been made. Regarding said manifestations in the analysis, we state that to perform a comprehensive descriptive analysis of the study population exceeds the scope of this study, which aims only at validating a novel causality algorithm. Moreover, the clinical and microbiological manifestations described are the ones that pertain to the algorithm itself as they constitute items within the algorithm (view Table in S3), such as autoimmune disease status, past history or concurrent disease of HHV-6. The literature supporting how these features are associated with DRESS is extense (DOI: 10.18176/jiaci.0480), hence their inclusion in the algorithm.
Retrospective Study Design , Sample Size and Methodology:
We acknowledge that our study is retrospective and based on a relatively small cohort of 37 patients with possible, probable, and definite DRESS syndrome. We understand your concern regarding the limitation of retrospective studies in providing a comprehensive analysis. To this extent the title of the manuscript has been adapted. And the methodology has been detailed further. Regarding sample size, DRESS is extremely low in prevalence and it is even more rare in cases where only one drug has been administered. It must be kept in mind that to validate the causality test (i.e., the ALDRESS algorithm) one must test it against both true positives and true negatives. Since there is no gold-standard for causality assessment in DRESS, we tested the algorithm and its most commonly applied counterpart (SEFV) against cases were the suspected culprit drug was the only drug administered, This is extremely rare to encounter, thus a prospective design would not be feasible.
Regarding bibliography:
We understand your concern about the age of references cited in our manuscript. However, we would like to point out that the majority of the references included in our study date from 2017 onwards. We acknowledge that one reference, the paper by Landis et al. from 1977, could appear outdated, however this publication pertains to the measurement of observer agreement rather than DRESS syndrome itself, a therefore is still relevant. We apologize for any confusion this may have caused and assure you that we have selected references that are relevant and provide important context for our study. We also appreciate your mention of the Journal of Clinical Medicine. However, to our knowledge, none of the papers cited in our manuscript have been published in the JCM journal. If there has been any misunderstanding or oversight in this regard, we apologize and would appreciate further clarification to address this matter appropriately.
Regarding minor changes to English: Text has been revised and improved to provide an easier comprehension and pacing. All minor typos detected have been corrected.
Regards,
Dr. S Stewart
Dr. E Ramírez

Reviewer 3 Report
Comments and Suggestions for Authors
Dear Authors,
I thoroughly read your work in form of a retrospective case-control study conducted to validate the ALDRESS algorithm - a standardized novel diagnostic tool for DRESS syndrome, as well as to compare it with SEFV algorithm (the current standard).
Here are my suggestions in form of the Minor revision:
- I could not understand the sentence within 2.2. section ("Controls included patients with more than one suspected was drug administered..."). Please rephrase it.
- The Results section should be improved, especially parts with Tables. It is expected for the results to be more clearly presented, in order to be easily understandable and followed by the readers. Current presentation of the results is not sufficient. Results were just undertaken from the statistical programme, and were not adjusted appropriately to the article format.
Sincerely,
Reviewer
Author Response
Dear Reviewer 3,
Thank you for your constructive feedback on our manuscript. We appreciate the opportunity to address your suggestions for minor revisions regarding clarity and presentation in the Results section.
Clarification of Sentence in 2.2 Section:
We apologize for the lack of clarity in the sentence within the 2.2 section. The sentence has been revised and rewritten to ensure its comprehensibility. Here is the revised version: "The control group in the studies comprised patients for whom more than one drug was under suspicion as a potential cause, with at least one of these drugs being identified as a culprit in the case group”.
Improvement of Results Section, Especially Tables:
We acknowledge your feedback regarding the presentation of results, particularly the tables, and agree that they should be more clearly presented for ease of understanding by readers. To this end, the following changes have been made to the manuscript:
- Clarity and Formatting: Clarity and formatting of the tables has been enhanced to ensure that the results are presented in a reader-friendly manner. This includes labelling, organization, and appropriate use of headings and subheadings. This now aligns with previous publications in the JMC.
- Interpretation: We understand the importance of providing meaningful interpretation alongside the results. We have ensured that the results are not merely presented from the statistical program but are also contextualized and explained within the framework of the article's objectives and hypotheses.
- Adjustment to Article Format: We have carefully reviewed the results to ensure that they are adjusted appropriately to adhere to the article format. This entailed restructuring or reorganizing the presentation to align with the overall flow of the manuscript.
Regards,
Dr. S Stewart
Dr. E Ramírez
